# Melatonin Reduces Alcohol Drinking in Rats with Disrupted Function of the Serotonergic System

**DOI:** 10.3390/jpm12030355

**Published:** 2022-02-26

**Authors:** Ieva Poceviciute, Rokas Buisas, Osvaldas Ruksenas, Valentina Vengeliene

**Affiliations:** Department of Neurobiology and Biophysics, Institute of Biosciences, Life Sciences Center, Vilnius University, LT-10257 Vilnius, Lithuania; ieva.poceviciute@gmc.vu.lt (I.P.); rokas.buisas@gf.vu.lt (R.B.); osvaldas.ruksenas@gf.vu.lt (O.R.)

**Keywords:** voluntary alcohol consumption, rats, escitalopram, serotonin, melatonin

## Abstract

The reason for the limited treatment success of substance-use-related problems may be a causal heterogeneity of this disorder that, at least partly, is manifested as differences in substance-use motives between individuals. The aim of the present study was to assess if rats with pharmacologically induced differences in the function of the serotonergic system would respond differently to melatonin treatment compared to control rats with respect to voluntary alcohol consumption. To achieve this goal, we treated rats neonatally with the selective serotonin transporter (SERT) inhibitor escitalopram. This procedure has been reported to cause long-lasting sleep abnormalities in rodents. The study demonstrated that during adulthood, rats that had been treated with escitalopram tended to drink higher amounts of alcohol compared to control rats. Further, administration of melatonin significantly decreased the alcohol intake in escitalopram-treated animals but caused only a slight, nonsignificant reduction in the alcohol consumption by control rats. In conclusion, our data support the therapeutic potential of melatonin as a treatment for alcohol use disorder. However, interindividual differences between alcohol users may considerably modify the outcome of the melatonin treatment, whereby patients that manifest lower sleep quality due to disruption of serotonergic activity are more likely to benefit from this treatment.

## 1. Introduction

The likelihood to initiate, maintain and lose control over substance use is influenced by a combination of multiple genetic/epigenetic factors and certain traumatic and lifestyle-related environmental experiences. These factors define not only personality traits and other characteristics of an individual (emotional, cognitive, motivational, social and other functions) but also personal (distinct) drinking motives. Accordingly, differences in drinking motives may be seen as a behavioural manifestation of the heterogeneity of substance use disorder, and may contribute to the limited treatment success of substance-use-related problems [1]. The aim of this study was to investigate if animals with pharmacologically induced differences in the function of the serotonergic system would respond differently to melatonin treatment with respect to voluntary alcohol consumption. Melatonin is a hormone that synchronises behaviours and physiological functions of the body to the presence/absence of daylight [2], and it has been demonstrated that compounds targeting the melatonergic system reduce drug-related behaviours in animals e.g., [3,4].

Disrupted function of the serotonergic system is achieved by exposing rodents to monoaminergic antidepressant treatment during early postnatal days (PD) for approximately 2 weeks [5,6,7]. It has been shown that adult animals, which had received neonatal administration of selective serotonin reuptake inhibitors (SSRIs), not only had reduced expression of SERT but also developed a similar phenotype to that seen in SERT knock-out animals [6,7]. The serotonergic system is responsible for many physiological functions, such as sensory processing, cognition, emotion regulation and sleep. SERT gene variants have also been associated with several psychiatric disorders including alcohol use disorder [8]. Thus, in adulthood, animals subjected to neonatal antidepressant treatment procedure have been shown to exhibit a number of behavioural impairments, including anxiety- and depression-like behaviour [7]. However, the most consistent phenotype shown is long-lasting sleep abnormalities, reported in numerous previous studies using neonatal antidepressant administration e.g., [5,6,9,10,11].

In the context of alcohol research, both animal and human studies identified low sleep quality as a drinking motive due to its association with increased probability of alcohol consumption [1,12]. For instance, a study of Currie at al. [13] showed that in half of the study’s participants, insomnia predated the onset of problematic alcohol use, and a study by Kuerbis et al. [14], using an ecological momentary assessment approach, demonstrated that poor sleep quality predicted increased alcohol drinking in young people. Similarly, animal studies showed that forced rapid eye movement (REM) sleep deprivation in outbred adult rats led to a transient increase in alcohol intake [15]. These findings suggested that targeting the body’s arousal and regulatory systems could be worth exploring when seeking novel strategies to treat alcohol use disorder [1,3,16].

In the present study, rat pups received once daily SSRI escitalopram administration between PD8 to PD21. In adulthood, rats were given free access to alcohol for several weeks. Thereafter, both control and escitalopram treated rat groups were exposed to repeated melatonin administration. Preference for sucrose and quinine solutions were tested in all animals to control for differences in reward- and/or taste-sensitivity.

## 2. Materials and Methods

### 2.1. Animals

A total of 24 1-week-old outbred female Wistar rats and 24 1-week-old outbred male Wistar rats (from our own breeding colony at the Vilnius University, Lithuania) were used. Animals were housed in groups throughout their adolescence into adulthood. Adult animals were housed individually in standard rat cages under a 12/12 h artificial light/dark cycle (lights on at 7:00 a.m.). Room temperature was kept constant (temperature: 22 ± 1 °C). Standard laboratory rat food (4RF21-GLP, Mucedola srl, Milan, Italy) and tap water were provided ad libitum throughout the experimental period. All experimental procedures were approved by the State Food and Veterinary Service of the Republic of Lithuania and were carried out in accordance with the local Animal Welfare Act and the European Communities Council Directive of 22 September 2010 (2010/63/EU).

### 2.2. Drugs

Escitalopram (AH diagnostics A/S, Denmark) was dissolved in 0.9% saline and injected as a volume of 3 mL/kg subcutaneously (SC). Melatonin (AH diagnostics A/S, Denmark) was suspended in water for injections containing 0.5% methylcellulose (Sigma-Aldrich, Merck KGaA, Darmstadt, Germany) and injected as a volume of 3 mL/kg intraperitoneally (IP). Control animals received an equal volume of respective vehicle.

### 2.3. Neonatal Escitalopram Treatment Procedure

All rats were divided into four groups and received a total of 14 (once daily) administrations of either saline (*n* = 12 and *n* = 12 for male and female rats, respectively) or 10 mg/kg of escitalopram [6] (*n* = 12 and *n* = 12 for male and female rats, respectively), which started on postnatal day 8 (PD8) and continued until PD21 [17,18]. From PD21 onwards, all animals were left undisturbed in their home-cages until adulthood, Figure 1.

### 2.4. Sucrose and Quinine Preference Tests

In order to identify differences in taste sensitivity, concentration of sucrose and quinine solutions were chosen according to our pilot studies to ensure that both tastes were just above detection threshold. For these tests, rats were separated into single cages and then given ad libitum access to two water bottles for 24 h (habituation phase). On the following day, one water bottle was substituted with 0.2% sucrose solution (*w*/*v*) for another 24 h. Thereafter, all rats were again given two water bottles and, 24 h later, one water bottle was substituted with a water bottle containing 0.03 mM quinine hydrochloride (Sigma-Aldrich, Merck KGaA, Darmstadt, Germany) for another 24 h. From these values, sucrose and quinine solution preference over water was calculated.

### 2.5. Voluntary Intermittent Alcohol Consumption

To measure the effects of neonatal escitalopram treatment on voluntary home-cage alcohol consumption, rats were given intermittent (every second day) access to tap water and alcohol solution for a total of 18 days. Rats subjected to intermittent alcohol access acquire higher levels of alcohol intake compared to rats that have continuous alcohol availability [19]. During this procedure, rats had access to 5% ethanol solution (*v*/*v*) for 3 days, 8% ethanol solution (*v*/*v*) for the next 3 days and 10% ethanol solution (*v*/*v*) for 3 more days. Water and alcohol intake were measured daily, and from these data, alcohol consumption (in g of pure ethanol/kg of body weight per day, g/kg/day) and water consumption (in ml/kg/day) were calculated.

### 2.6. Melatonin Treatment

To examine the effects of melatonin on voluntary alcohol consumption, 12 control and 12 escitalopram-treated male rats were given continuous access to two bottles, one containing tap water and another 10% ethanol solution, for 1–2 weeks until stable baseline drinking was established. Before the pharmacological study, stable baseline drinking was monitored for at least 4 days. After the last day of baseline measurement, each animal was subjected to a total of three injections (once daily, at 6:00–6:30 p.m.) of either vehicle or 40 mg/kg of melatonin (the dose was chosen based on our previous studies [3,4]). Thereafter, animals were left undisturbed until baseline drinking levels recovered and the procedure was repeated one more time using a Latin square design until vehicle and melatonin were tested in all animals. Animals that reduced voluntary baseline alcohol intake below 1 g/kg/day were excluded from the study to avoid the ‘floor’ effect (i.e., 2 control rats were excluded before the first treatment round and 4 escitalopram-treated rats did not recover alcohol intake after the first treatment round). Bottle weight was recorded daily, and alcohol consumption (in g/kg/day) and water consumption (in ml/kg/day) were calculated.

### 2.7. Statistics and Data Analysis

Data obtained from sucrose and quinine preference tests and weekly body weight measurements were analysed by independent two-tailed t-test (factor–group, i.e., control vs. escitalopram treated). Data derived from baseline home-cage drinking (alcohol and water intake) was analysed using a two-way analysis of variance (ANOVA) with repeated measures (factors were: group and day). Data analysis regarding the effects of treatment on the change in the animals’ body weight (measured before and after melatonin treatment) was performed using a two-way ANOVA (factors were: group and treatment). Whenever significant differences were found, post hoc Student Newman–Keuls tests were performed. The chosen level of significance was *p* < 0.05.

## 3. Results

### 3.1. Neonatal Escitalopram Treatment Procedure

No animal died due to neonatal treatment procedure. However, escitalopram treatment induced a growth retardation in both female and male rats. Hence, at the start of the behavioural tests, both female and male rats exhibited significantly lower body weight by approximately 7%) compared to their control counterparts (factor treatment group: t_(22)_ = 3.1, *p* < 0.01 and t_(22)_ = 2.3, *p* < 0.05 for female and male rats, respectively). Once of adult age, male rats slowly recovered their body weight so as during the voluntary alcohol drinking procedure this difference was no longer significant from control rats (*p* = 0.09 and *p* = 0.44 at the beginning and at the end of voluntary intermittent drinking, respectively). Female rats, however, remained smaller throughout the entire intermittent alcohol consumption study, and hence these rats were not used in the melatonin-treatment experiment.

### 3.2. Sucrose and Quinine Preference Tests

Data analysis revealed that escitalopram treated rats were not significantly different from control rats in reward- and/or taste-sensitivity. Thus, all animal groups had almost equal preference (i.e., approximately 72%) for 0.2% sucrose solution over water during the 24 h free-choice sucrose preference test (*p* = 0.52 and *p* = 0.82 for female and male rats, respectively) (data not shown). Similarly, no significant difference was found between groups during the 24 h free-choice quinine preference test (quinine preference was approximately 20%) (*p* = 0.68 and *p* = 0.82 for female and male rats, respectively) (data not shown).

### 3.3. Voluntary Intermittent Alcohol Consumption

Analysis of voluntary intermittent alcohol drinking data revealed that alcohol and water intake changed over time in both female (factor day: F_(8,215)_ = 5.9, *p* < 0.001 and F_(8,215)_ = 14.3, *p* < 0.001 for alcohol and water, respectively) and male (factor day: F_(8,215)_ = 7.6, *p* < 0.001 and F_(8,215)_ = 16.8, *p* < 0.001 for alcohol and water, respectively) rats (Figure 2). These changes were most apparent for water intake and were related to increasing alcohol concentration. No difference in either alcohol or water consumption was found between control and escitalopram treatment in both female (factor group: *p* = 0.39 and *p* = 0.76 for alcohol and water, respectively) (Figure 2A,C) and male (factor group: *p* = 0.14 and *p* = 0.61 for alcohol and water, respectively) (Figure 2B,D) animal groups.

### 3.4. Melatonin Treatment

The overall data analysis using a two-way repeated measures ANOVA indicated that treatment with melatonin caused significant changes in alcohol intake in male rats (factor day: F_(15,287)_ = 5.3, *p* < 0.001). However, a significant reduction in basal alcohol intake was observed only in escitalopram-treated rats (factor group × day interaction effect: F_(15,287)_ = 1.7, *p* = 0.05) (Figure 3). Hence, post hoc analysis of alcohol intake revealed that control animals tended to consume lower amounts of alcohol during melatonin treatment compared to vehicle treatment days (Figure 3A), whereas significant reduction of alcohol intake (by approximately 30%) was seen in escitalopram treated rats (Figure 3B). Water intake was either unchanged or slightly increased during melatonin treatment days (factor day: F_(15,287)_ = 2.1, *p* < 0.01 and factor group × day interaction effect: *p* = 0.37) (data not shown). The opposite effect of melatonin treatment on water consumption demonstrates selectivity of treatment towards alcohol. It should be mentioned, however, that treatment of rats with melatonin caused small (−1%) but significant loss of body weight in both control and escitalopram treated animal groups (factor treatment: F_(1,35)_ = 27.1, *p* < 0.001 and factor group × day interaction effect: *p* = 0.12), indicating that food intake and/or metabolism was affected by this dose.

## 4. Discussion

The present study demonstrated that adult outbred rats treated with selective serotonin reuptake inhibitor escitalopram during early postnatal weeks, had similar reward- and/or taste-sensitivity as control rats, but tended to drink higher amounts of alcohol. Repeated administration of melatonin before the onset of the dark phase caused a clear but not significant reduction in voluntary baseline home-cage alcohol consumption by control rats and a significant decrease in alcohol intake by rats treated with escitalopram.

Several earlier studies demonstrated that neonatal antidepressant administration led to an increased free-choice alcohol consumption in adult animals [5,17,18,20]. However, the study of Hilakivi and Sinclair [20] revealed that administration of the antidepressant clomipramine during early postnatal weeks increased the voluntary alcohol consumption in alcohol-preferring AA (Alko, Alcohol) rats but not in alcohol-avoiding ANA (Alko, Non-Alcohol) rats [20]. This suggests that besides serotonergic activity, other factors, such as genetic background of an animal, may have a significant impact on the development of alcohol-drinking behaviour. In the present study, neonatal escitalopram treatment did not cause a significant increase in alcohol consumption. Interestingly, SERT knockout mice have also been shown to consume similar amounts of alcohol as their wild-type counterpart mice [21] confirming that alterations in serotonergic activity may not be sufficient to cause an increase in alcohol consumption.

Despite lack of significant impact of neonatal escitalopram administration on alcohol consumption, the present study showed that this procedure had a significant impact on the efficacy of melatonin with regard to reducing voluntary alcohol intake. In our earlier study, we demonstrated that activation of the melatonergic system just before the onset of the dark phase induced a circadian phase advance and reduced relapse-like drinking in outbred male rats [3]. The study also suggested that restoring normal sleep architecture had the additional beneficial effect of reducing alcohol consumption [3]. The present study extends these findings by demonstrating that the response to melatonin may depend on individual characteristics of animal with melatonin treatment being more effective in animals with disrupted function of the serotonergic system. As mentioned above, this procedure is known to cause long-lasting sleep alterations and other behavioural changes in animals e.g., [5,6,9,10,11]. These findings are in line with previous studies showing that individual differences may have a considerable impact on the treatment outcome. For instance, our earlier study demonstrated that nalmefene was most effective in rats that consumed greater amounts of highly concentrated alcohol per drinking approach [22]. The impact of individual differences has also been demonstrated in human studies. The opioid receptor antagonist naltrexone reduced craving during nondrinking moments and the likelihood of heavy drinking specifically in carriers of the dopamine receptor DRD4-L allele variant [23], whereas 5-HT3 receptor antagonist ondansetron was more effective at reducing the number of drinks per day in individuals with L/L polymorphism in the promoter region of SERT [24].

In conclusion, our data support the therapeutic potential of melatonin as a treatment for alcohol-use disorder. However, interindividual differences between alcohol users may considerably modify the outcome of melatonin treatment, where patients that manifest lower sleep quality due to disruption of serotonergic activity are more likely to benefit from this treatment.

The efficacy of melatonin may be explored further by use of animals with genetic alterations of SERT function.

## Figures and Tables

**Figure 1 jpm-12-00355-f001:**
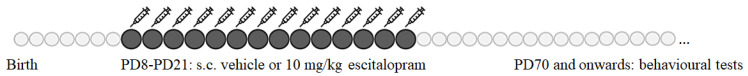
All rats were divided into four groups and received a total of 14 (once daily) s.c. administrations of either saline (vehicle, *n* = 12 and *n* = 12 for male and female rats, respectively) or 10 mg/kg of escitalopram (*n* = 12 and *n* = 12 for male and female rats, respectively), which started on postnatal day 8 (PD8) and continued until PD21 (treatment days are marked as dark circles). From PD21 until adulthood (PD70) all animals were left undisturbed in their home-cages (marked as white circles). From PD70 onwards, the following behavioural tests were performed: sucrose and quinine preference tests, voluntary intermittent alcohol consumption and effect of melatonin administration on voluntary alcohol consumption (please note that escitalopram-treated female rats remained smaller than their control counterpart rats throughout the sucrose and quinine preference tests and the entire intermittent alcohol consumption study, and hence these rats were not used in the final melatonin treatment experiment).

**Figure 2 jpm-12-00355-f002:**
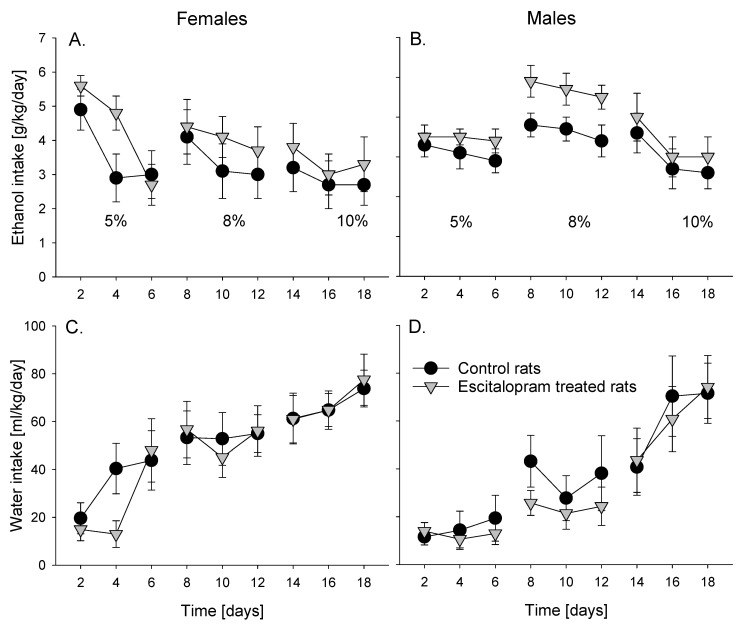
Total daily ethanol (in g of pure ethanol per kg of body weight per 24 h, g/kg/day, (**A**,**B**)) and water (in ml/kg/day, (**C**,**D**)) intake in control and escitalopram-treated female (**A**,**C**) and male (**B**,**D**) rats (*n* = 12 per group). All rats were given intermittent (every second day) access to tap water and alcohol solution for a total of 18 days. During this procedure, rats had access to 5% ethanol solution for 3 days, 8% ethanol solution for the next 3 days and 10% ethanol solution for 3 more days. Data are presented as means ± S.E.M.

**Figure 3 jpm-12-00355-f003:**
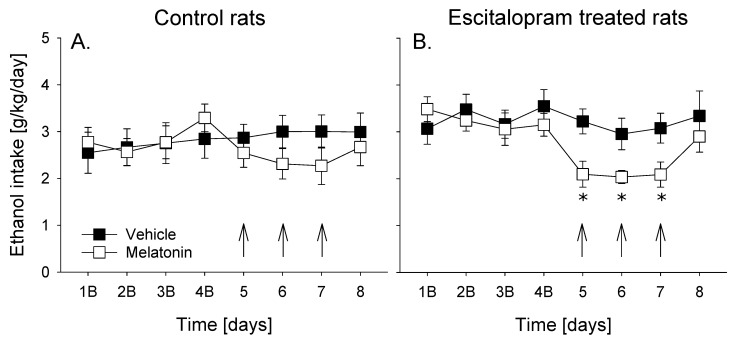
Total daily ethanol (in g of pure ethanol per kg of body weight per 24 h, g/kg/day, (**A**,**B**)) intake in control ((**A**), *n* = 10) and escitalopram treated ((**B**), *n* = 8) male rats before (baseline, 1B, 2B, 3B and 4B), during (arrows) and after administration of either vehicle or 40 mg/kg of melatonin. Drug administration was performed once daily during three consecutive days at the end of the light phase. Data are presented as means ± S.E.M. * indicates significant differences from vehicle treatment, p < 0.05.

## Data Availability

The data underlying this article will be shared on reasonable request to the corresponding author.

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
