# Peer review of "Melatonin Reduces Alcohol Drinking in Rats with Disrupted Function of the Serotonergic System"

_jpm, 2022, doi:10.3390/jpm12030355_

Round 1
Reviewer 1 Report
The paper describes the effect of melatonin intake on alcohol consumption. The authors reasonably obtained the results from the proposed experimental design. However, the experiment is somewhat complicated; thus, the reviewer suggests providing the figure of experimental design in the text, if possible.
Author Response
Dear Editor,
We truly appreciated the comments of the reviewers, which gave us the opportunity to improve the quality of the manuscript. Point by point responses to reviewers’ comments are listed below.
1. Comment: the experiment is somewhat complicated; thus, the reviewer suggests providing the figure of experimental design in the text, if possible.
Reply: The figure with the experimental design has been added. Please see the attachment.
2. Comment: Moderate English changes required
Reply: The manuscript has been provided to an external reviewer for language corrections.

Reviewer 2 Report
In the attached PDF, you will find some notes.
I think you could improve the design of your experiments to find robust results and detect significant changes. i.e., increasing the day of intake and administration increase the number of experimental subjects. I suggest you consult article about this.
Their results do not support the conclusions forcefully.

Author Response
Dear Editor,
We truly appreciated the comments of the reviewers, which gave us the opportunity to improve the quality of the manuscript. Point by point responses to reviewers’ comments are listed below. Please also see the attachment.
1. Comment: I think you could improve the design of your experiments to find robust results and detect significant changes. i.e., increasing the day of intake and administration increase the number of experimental subjects. I suggest you consult article about this.
Reply: Combining data sets from two studies is considered as a form of p-hacking (i.e., a practice where data is manipulated until it reaches significance). Estimation of the experimental group size was carried out before the study using previous data from the similar experiments. Variability of individual data in the present study confirmed that adequate group size was chosen.
2. Comment: Their results do not support the conclusions forcefully.
Reply: Multiple earlier publications demonstrated that neonatal administration of escitalopram has long-lasting impact on animal behaviour, including sleep alterations. The present study demonstrated that two groups of rats (control vs escitalopram treated) responded differently to melatonin treatment. Our conclusion, “inter-individual differences between alcohol users may considerably modify the outcome of the melatonin treatment”, suggests that there is a possibility that some alcohol users might respond differently to melatonin treatment than others. Ideally, the study could have measured sleep alterations in escitalopram treated rats to support the conclusion of the study. However, it has been demonstrated multiple times in the earlier studies and this is the reason why the present study is submitted as brief communication.
3. Comment: P4, do not put abbreviations, if you have not defined them.
Reply: This abbreviation (i.e., PD) is defined in the first sentence of the 2nd paragraph of the Introduction.
4. Comment: P4, Remove this sentence from here. It does not an aim of the study; put it in another place, please you.
Reply: The sentence has been moved to the first paragraph.
5. Comment: P5, delete.
Reply: Sentence on body weight measurements has been deleted from the text.
6. Comment: P5, A single daily administration? to clarify this. for a better understanding.
Reply: "daily" has been changed to "once daily"
7. Comment: P5, Reference? put a reference about this protocol.
Reply: References for the protocol have been added:
- Popa D, Léna C, Alexandre C, Adrien J. Lasting syndrome of depression produced by reduction in serotonin uptake during postnatal development: evidence from sleep, stress, and behavior. J Neurosci. 2008, 28, 3546-3554
- Hilakivi LA, Sinclair JD, Hilakivi IT. Effects of neonatal treatment with clomipramine on adult ethanol related behavior in the rat. Brain Res. 1984, 317, 129-132
- Neonatal clomipramine treatment, alcohol intake and circadian rhythms in rats. Psychopharmacology. 1998, 138, 176-183
8. Comment: P6, put a reference about this protocol.
Reply: Measurement of preference for one solution over the other is a simple test used by virtually all behavioural laboratories. The reference to this protocol may be unnecessary.
9. Comment: P6, protocol reference?
Reply: This drinking schedule is used by many laboratories in alcohol research field. We can, however, reference Wise (1973) since this was one of the first publications that showed benefits of this drinking schedule in rats: Wise RA. Voluntary ethanol intake in rats following exposure to ethanol on various schedules. Psychopharmacologia. 1973;29(3):203-210. doi:10.1007/BF00414034
10. Comment: P8, The trend is not relevant if the statistical test does not indicate that the change is significant.
Reply: This sentence has been excluded from the text.
11. Comment: P10, ?
Reply: AA and ANA are the names of two well-known rat lines. The origin of the name is Alko, Alcohol for AA and Alko, Non-Alcohol for ANA. This specification of the origin is now added to the Discussion.
12. Comment: P15, delete “pure”.
Reply: “Pure ethanol” is used to make clear that y-axis shows intake of pure ethanol, not intake of 10% ethanol solution. Showing intake of pure ethanol is used in alcohol research so that the readers could compare ethanol intake between different studies, since different labs use different dilutions of ethanol in their experiments.
13. Comment: P16, the dose of melatonin used is very high, could you explain your selection criteria for this dose?
Reply: The dose was chosen based on our previous studies: Vengeliene et al., 2015 and Takahashi et al., 2017. The discussion on the choice of a dose is provided in Vengeliene et al 2015. Reference to both publications is provided in the Method section.
